

# A new phase in the production of quality-controlled sea level data

Graham D. Quartly[1], Jean-François Legeais[2], Michaël Ablain[2], Lionel Zawadzki[2], M. Joana Fernandes[3], Sergei Rudenko[4,5], Loren Carrère[2], Pablo Nilo García[6], Paolo Cipollini[7], Ole Baltazar Andersen[8], Jean-Christophe Poisson[2], Sabrina Mbajon Njiche[9], Anny Cazenave[10,11], Jérôme Benveniste[12]

[1]Plymouth Marine Laboratory, Plymouth, PL1 3DH, United Kingdom
[2]CLS, 31520 Ramonville-Saint-Agne, France
[3]University of Porto, 4099-002 Porto, Portugal
[4]Deutsches Geodätisches Forschungsinstitut, Technische Universität München, 80333 Munich, Germany
[5]Helmholtz Centre Potsdam GFZ German Research Centre for Geosciences, Telegrafenberg 14473 Potsdam, Germany
[6]isardSAT, 08042 Barcelona, Spain
[7]National Oceanography Centre, Southampton, SO14 3ZH, United Kingdom
[8]DTU Space, 2800 Kongens Lyngby, Denmark
[9]CGI, Leatherhead, KT22 7LP, United Kingdom
[10]LEGOS, 31400 Toulouse, France
[11]ISSI, 3912 Bern, Switzerland
[12]ESA/ESRIN, 00044 Frascati, Italy

*Correspondence to*: Graham D. Quartly (gqu@pml.ac.uk)

**Abstract.** Sea level is an Essential Climate Variable (ECV) that has a direct effect on many people through inundations of
coastal areas, and it is also a clear indicator of climate changes due to external forcing factors and internal climate variability.
Regional patterns of sea level change inform us on ocean circulation variations in response to natural climate modes such as
El Niño and the Pacific Decadal Oscillation, and anthropogenic forcing.  Comparing numerical climate models to a
consistent set of observations enables us to assess the performance of these models and help us to understand and predict
these phenomena, and thereby alleviate some of the environmental conditions associated with them. All such studies rely on
the existence of long-term consistent high accuracy datasets of sea level. The Climate Change Initiative (CCI) of the
European Space Agency was established in 2010 to provide improved time series of some ECVs, including sea level, with
the purpose of providing such data openly to all to enable the widest possible utilisation of such data. Now in its second
phase, the Sea Level CCI project merges data from 9 different altimeter missions in a clear and well-documented manner,
selecting the most appropriate satellite orbits and geophysical corrections in order to reduce the error budget. This paper
summarises the corrections required, the provenance of corrections and the evaluation of options that have been adopted for
the recently released v2.0 dataset (DOI: 10.5270/esa-sea_level_cci-1993_2015-v_2.0-201612). This information enables
scientists and other users to clearly understand which corrections have been applied and their effects on the sea level dataset.
However, the overall result of these changes is that the rate of rise of global mean sea level still equates to ~3.2 mm yr$^{-1}$
during 1992-2015.

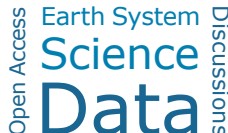

## 1 Introduction

Sea level is widely recognised as an Essential Climate Variable (ECV) that has a significant impact on mankind. An accelerated rise in global mean sea level (GMSL) shows the integrated effect of increased ocean heat content and the enhanced melting of glaciers and ice sheets. Many major conurbations are sited on the coast and vulnerable to long-term sea
level rise. This is also critical for low-lying islands (such as the Maldives) and highly populated river deltas (such as the Brahmaputra in Bangladesh) where continued sea level rise threatens the lives of many.

The issue of sea level rise thus has aspects that are global, regional and local. Global mean sea level rise is related to increased forcing within the global climate through increased ocean warming and land ice loss. Satellite altimetry also reveals significant regional variability, with some regions experiencing greater rates of sea level rise. The sea level in a
region also responds to decadal changes associated with various climatic phenomena, which may temporally ameliorate or exacerbate the effects of global change; such regional climatic oscillations need to be better measured, modelled and understood. An accurate robust record of regional changes can help provide the "fingerprint" to distinguish between different models of the Earth's response to enhanced climate forcing. At the coast what is important to the population is the combined effects of large-scale climate variations, local changes in waves and currents, and vertical land motion. In many
regions the ground is subsiding in response to increased sediment load in deltas or ground water depletion near megacities. Together ground subsidence and sea level rise amplify the vulnerability of coastal regions, producing major societal impacts. Finally, sea level variations need to be precisely monitored at the mesoscale, (50-200 km) as the variability associated with eddies and current fluctuations provide many of the mechanisms for transporting and mixing water masses, with attendant effects on primary productivity.

The European Space Agency (ESA) set up the Climate Change Initiative (CCI) in 2010 to develop consistent long-term datasets of many of the recognised Essential Climate Variables (ECVs), with one using satellite altimetry to provide sea level data over most of the open ocean, with the aim of addressing part of the afore-mentioned wide range of scientific and societal needs. The initial (v1) dataset spanned 1993-2010 (Ablain et al., 2015; Ablain et al., 2016); the second phase of the CCI (2014-2016) has not only extended the data duration (up to end of 2015), but also revisited many aspects of the data
processing and corrections to improve the quality of the dataset for global, regional and mesoscale applications. This paper details the processing options selected for the production of the v2.0 dataset.

The whole dataset is based on the concept of altimetry i.e. that a satellite flying in a near-polar orbit measures the ocean surface topography by recording the time taken for radar pulses emitted by the satellite to reflect off the surface and be recorded on the satellite. There are many technical details to the measurement of this distance to within a few centimetres
from a satellite ~800-1350 km above the Earth's surface, which are described in Chelton et al (1989), Fu and Cazenave (2001) and Escudier et al. (2017). Range is then computed by multiplying the time delay by the speed of light *in vacuo*, and then applying corrections for the components of the return path where speed is slightly less — these are the dry tropospheric correction (DTC), wet tropospheric correction (WTC), and the ionospheric correction (Iono). Subtracting this altimetric





range from a well-modelled orbit height then gives a value for the sea surface height relative to some reference surface. To give a measure that is useful for oceanographic applications, the value needs to be adjusted for the effect of changes in atmospheric conditions (Dynamic Atmosphere Correction, DAC) and tides. Finally there is an empirical correction, sea state bias (SSB), accounting for various effects related to the wind and wave conditions. Thus the required oceanographic

parameter, the sea level anomaly (SLA), is defined as:

$$SLA = Orbit - (Range + DTC + WTC + Iono) - DAC - Tides - SSB - MSS \qquad (1)$$

where the mean sea surface (MSS) is the sum of the geoid (the geopotential surface indicating the level that would be recorded for a  motionless ocean) and the mean dynamic topography (MDT), which corresponds to the topographic variations associated with the mean circulation of the ocean. Values for these corrections are supplied in the geophysical

data records (GDR) provided by the space and meteorological agencies; however there is a need to review whether new ones are more accurate, and also to establish a consistent selection across all missions used.

In progressing from the Sea Level CCI (SL_cci) v1.1 product to v2.0, the length of the dataset has been extended and two new sources of altimeter data have been included (SARAL/AltiKa and CryoSat-2, see Fig. 1), and all the corrections have been reappraised to ensure that they are the most appropriate for establishing a consistent and stable long-term record for use

at global, regional and meso- scales. Note that the SL_cci Algorithms Theoretical Basis Document (Ablain et al., 2016) provides the details on all algorithms used to compute the 1 Hz along-track measurements. This paper deals with each of these correction terms, documenting the selections made and their justification; subsequent papers will exploit the SL_cci v2 data to improve our understanding of present-day sea level variations at global and regional scales, and their causes.

The assessment of new corrections has been carried out by a formal validation protocol using a common set of diagnoses

defined to fulfill the sea level accuracy and precision requirements, as defined by the Global Climate Observing System (GCOS, 2011). This protocol consists of comparing new altimeter corrections with previous ones through their impact on the sea level calculation. The validation diagnoses are distributed into three distinct families allowing the assessment of altimetry data with complementary objectives.

1. "Global internal analyses" which check the internal consistency of a specific mission-related altimetry system by

analyzing the computed sea level, its instrumental parameters (from altimeter and radiometer) and associated geophysical corrections,

2. "Global multi-mission comparisons" which evaluate the coherence between two different altimetry systems through comparison of SLA data,

3. "Altimetry comparison with in situ data" which computes differences between altimeter SLA data and those from

*in situ* sea level measurements, e.g. tide gauges or Argo-based steric sea level data (Legeais et al., 2016); this third approach allows for the detection of potential drifts or jumps in the long-term sea level time series.





## 2 Orbits and range

Orbital height and altimeter range are the two large terms that are differenced in the calculation of SLA. The former term refers to the height of the satellite above the reference ellipsoid, whilst the range is the measurement from the radar altimeter. The orbit is not measured everywhere, but calculated from a sophisticated numerical theory of satellite motion using a well

defined reference frame and taking into account various forces acting on a satellite, such as gravitational fields of the Earth, Moon, Sun and major planets of the Solar System, drag in the Earth's atmosphere, and radiation from the Sun and the Earth. Precise satellite laser ranging from ground stations, GNSS locations from navigation satellites that are in a much higher orbit, and radio-positioning information from DORIS and PRARE are used in the orbit computations for the various altimetry satellites. The calculation of altimeter range includes waveform retracking and compensation for an altimeter bias

specific to the instrument (Ablain et al., 2017, Escudier et al., 2017).

### 2.1 Modelled orbits

As the orbital height of the satellites needs to be known to centimetric accuracy (i.e. one part in $10^8$), the Earth's gravity field requires a detailed representation usually expressed in spherical harmonic coefficients, typically to degree and order 90-120 for satellites at altitudes between 700 and 1400 km. Terrestrial gravimeters and geodetic satellites, such as LAGEOS, and

more recently the space gravimetry mission GRACE (Tapley et al., 2004) revealed that the Earth's gravity field changes with time. Detailed analysis of the observations of satellites in low Earth orbit, in particular, from the missions designed to observe the Earth's gravity field, such as CHAMP (2000-2010), GRACE (2002-present) and GOCE (2009-2013) significantly improved knowledge about Earth's static and time-variable gravity including mass redistribution in the Earth's atmosphere, hydrosphere, ocean and cryosphere. This concept, referred to as time-varying gravity, accepts that the spherical

coefficients of the gravity field will change both seasonally and with epoch. Ollivier et al. (2012) and Rudenko et al. (2014) showed that ignoring a time-variable (secular) part of the geopotential causes up to 3 mm yr$^{-1}$ east-west errors in the regional sea level trends. Additionally, ignoring non-tidal high-frequency atmospheric and oceanic mass variations can lead to errors of up to 7 mm in sea level height and up to 0.25 mm yr$^{-1}$ in the regional trend (Rudenko et al., 2016). To achieve precise orbits also requires an accurate model of the spacecraft itself, in order to understand the drag terms from a very tenuous

atmosphere, the effects of solar radiation pressure and to take into account relativistic effects.

New VER11 orbit solutions of ERS-1, ERS-2, Envisat, TOPEX/Poseidon, Jason-1 and Jason-2 have been generated at GFZ (Rudenko et al., 2017). Additionally, a new orbit version (POE-E) has been computed at CNES for Jason-1, Jason-2, AltiKa and CryoSat-2, and finally, a new orbit version (GSFC std1504) has been derived at GSFC for TOPEX/Poseidon, Jason-1 and Jason-2. All these orbit solutions have been derived in the extended ITRF2008 reference frame (Altamimi et al., 2011)

by using SLRF2008 (Pavlis et al., 2009), DPOD2008 (Willis et al., 2016) and IGS08 (Rebischung et al., 2012) station solutions and are based on the GDR-E orbit standards (OSTM, 2015) or similar standards. The main differences of the GDR-E orbit standards with respect to the previous GDR-D (OSTM, 2015) orbit standards consist of i) using a more refined Earth



time-variable gravity field model EIGEN-GRGS.RL03-v2.MEAN-FIELD including time-variable geopotential terms up to degree and order 80 (instead of 50 in the previous standards), ii) increased expansion of the atmospheric gravity model (from degree and order 20 to 70), iii) modelling of tidal and non-tidal geocentre variations, iv) improved modelling of non-gravitational forces for some satellites, v) improvements in the troposphere correction model for DORIS observations, and

vi) using Earth orientation parameters consistent with the ITRF2008 reference frame.

A validation of these new orbit solutions has been performed with respect to those selected for the SL_cci v1.0 product (Ablain et al., 2015, Table 1). The main criteria for the selection are a reduction of the SLA crossover variance differences and minimum absolute difference of the MSL computed using ascending and descending passes. As a result of this validation, the following orbits have been selected: GFZ VER11 orbits for ERS-1, ERS-2 and Envisat, CNES POE-E orbits

for Jason-1, Jason-2, AltiKa and CryoSat-2 and GSFC std1504 orbit for TOPEX/Poseidon. Since no new orbit solution has become available for GFO, the same (GSFC std08) orbit was used for the generation of the SL_cci v2.0 product, as for its predecessor. Couhert et al. (2015) showed that using Jason-1/2 orbits derived with SLR and DORIS measurements may cause up to 0.3 mm yr$^{-1}$ decadal and 1 mm yr$^{-1}$ interannual regional errors, when employing ITRF2005 reference frame instead of ITRF2008 one for orbit computations. Since no DORIS data were used to derive GFO GSFC std08 orbit, the

impact of using this orbit on the regional sea level may be lager, when using just one mission. However, since regional sea level is derived in the SL_cci v2.0 product using data from nine altimetry missions over the time span 1993-2015, the impact of using GFO orbit derived in ITRF2005, while the orbits of other eight missions are in the ITRF2008, is rather small. There is no impact of the GFO orbit on the global mean sea level (GMSL), since GFO is not included in the reference missions used to derive that in the SL_cci v2.0 product.

The SL_cci v1 product used the REAPER combined orbit for ERS-1 and ERS-2 (Rudenko et al., 2012), whilst GFZ VER11 orbit was used for the new (v2) product detailed here. Figure 2 shows the differences in sea level trends computed using two these different orbits. Another example is the changes in Jason-1 trends engendered mission by the new orbit standards (Fig. 3). The broad dipole pattern corresponds to the modelling of geocentre motion, whilst individual tracks are prominent where changes to the gravity field have a more local effect.

**2.2 Precise determination of the altimeter range**

A waveform i.e. the full radar echo recorded on-board the altimeter corresponds to a disk a few kilometres across on the sea surface. Provided the surface is homogeneous, the shape of the waveform will conform to the Brown model (Brown, 1977; Hayne, 1980). In such circumstances, the position of the waveform (and thus the range) may be very accurately extracted; these values are stored in the geophysical data records (GDR) provided by the space agencies. In general, the sea level CCI

project has not attempted to perform its own retracking of all the different missions, but assessed the quality of those available. In particular, the v2.0 product makes use of the latest ERS-1 and ERS-2 reprocessings from the REAPER project (Brockley et al., 2017), and incorporates the new GDR (version E) for Jason-1, which includes improved estimates of





internal errors. However, although there are known to be artefacts in the TOPEX waveform data, no new product for that mission is available in time for the reprocessed SL_cci v2.0 product.

As part of the Level 1b processing, corrections are applied to the range for changes in the Point Target Response (PTR) in response to ageing of the instrument, and also any drift in the Ultra-Stable Oscillator (USO) that controls the on-board timing

of pulses. Within the early years of the SL_cci project it had been found that Envisat's PTR waveform needed to be reversed in the Level 1b processing at Ku-band (García and Roca, 2010); this change caused a notable impact on range, leading to a better agreement of the long-term trends between Envisat and the reference missions. During the second phase, the S-band signal was assessed, but no change was made because there was no discernible benefit.

During the first phase of the SL_cci project, the coastal zone and the Arctic had been recognised as two areas requiring

special effort because the waveforms were not "Brown-like" due to inhomogeneities within the full instrument footprint. Waveforms in coastal regions may contain early contributions from land or "bright target" responses from glassy seas in sheltered regions (Gómez-Enrí et al., 2010). The SL_cci project has been assessing two methodologies to overcome such anomalous waveforms: including a Gaussian peak within the shape model (Halimi et al., 2013) or focussing the shape-fitting only on the leading edge (Passaro et al., 2014). In the Arctic, the inhomogeneities are due to a mix of ice floes and thin leads

(gaps within the ice exposing very calm waters). Poisson et al. (2017) have developed a processing scheme for classifying the data according to reflecting surface and retracking the waveforms from leads using an extended Brown model. So far, only data from the Envisat and SARAL/AltiKa missions have been processed which has led to the production of a promising Arctic sea level product now available for the users. However, both the coastal and Arctic work are part of ongoing research, and additional efforts are required so that these retracked data could be included in a future SL_cci product.

**3 Corrections to atmospheric propagation**

The main atmospheric retardation of the radar signal, the dry tropospheric correction (DTC), is simply due to the mass of neutral dry air that it propagates through, and that can be retrieved from atmospheric pressure at sea level. As that cannot be measured from space, what is required is a good atmospheric model that incorporates measurements i.e. a reanalysis product. The wet tropospheric component (WTC), representing the extra delay from atmospheric water vapour and liquid water, can

also be extracted from an assimilating model, but the scales of temporal and spatial variations of the water vapour are usually not adequately resolved by global reanalyses, so some direct measurement of water vapour and liquid water is beneficial. Most altimetric satellites carry a nadir-viewing microwave radiometer (MWR) to record relevant emissions for WTC retrieval; however CryoSat-2 has no such package, as its focus is on polar latitudes where the WTC may largely be neglected. However microwave radiometers are not reliable in the coastal zone due to their large footprint (typically 20-40

k) and global atmospheric models lack the resolution to incorporate coastal processes. An alternative data source is provided by shore-based GNSS stations, as the WTC derived from their L-band measurements is also valid at Ku- and Ka-band, since the troposphere is a non-dispersive medium at these frequencies.



The ionospheric delay is a retardation of the passage of radio waves by free electrons, which get accelerated. Such an effect predominantly occurs on the sun-facing side of the Earth, and is strongest in two bands near the tropics. It is proportional to the columnar total electron content (TEC) divided by the square of the radar frequency. The TOPEX, Jason and Envisat spacecraft were designed with dual-frequency altimeters specifically to allow an estimation of the pertinent ionospheric

correction from the difference in range delay recorded at the two frequencies. This was because the early ionospheric models were not deemed to be accurate enough to support the high precision required from the reference missions. However, there have been marked improvements in the ionospheric models in the past decade. Since AltiKa operates at Ka-band, the size of this correction is only one sixth of that for the other instruments (which operate at Ku) and so operation at multiple frequencies was not justified.

**3.1 Dry tropospheric correction**

Dry tropospheric corrections were calculated (Ablain et al., 2016) according to 3 different numerical models: ECMWF operational, ERA-Interim and JRA-55. Analysis of the sea level variance at crossovers and investigation of trends were performed for sea level data computed with each correction. Although the operational version of the ECMWF model has the highest spatial resolution for recent years, giving it a superior performance to the others, it is not consistent for the whole

20+ year period. The JRA-55 set of corrections led to greater mesoscale signal than ERA-Interim, especially at southern latitudes. Thus, from the long period reanalyses, the ERA-Interim corrections were the ones adopted for SL_cci v2.0.

**3.2 Wet tropospheric correction**

The University of Porto has developed a robust method for determining the WTC by data combination through space-time objective analysis of various data types: valid measurements from the on-board MWR (whenever available) and third party

observations from GNSS and scanning imaging MWR. The latest version of these corrections, designated GNSS-derived Path Delay Plus (GPD+, see Fernandes and Lázaro. 2016), includes improved calibration of all radiometers on altimetric satellites, by comparing them with the known stable performance of the SSM/I and SSM/IS. Adjustments are made first to the reference missions in 10-day repeat non-sun-synchronous orbits and then to the sun-synchronous 35-day missions. In addition to the calibration with respect to SSM/I and SSM/IS, the original GPD solution (Fernandes et al., 2015) has been

augmented by adding new datasets (from scanning imaging radiometers) and improved selection criteria for selecting valid MWR observations. The GPD+ correction is implemented in SL_cci v2.0 for all missions except GFO, although similar corrections have subsequently become available for this satellite (Fernandes and Lázaro. 2016). In SL_cci v2.0, the WTC for GFO is calculated from its MWR for observations located >50 km from the coast, and from the ECMWF operational model for data between 10 and 50 km from coast. There were problems with the radiometer during GFO cycles 135-137,

166, 181, 189 and after 201; in such cases ECMWF values were used for all observations.

The GPD+ correction allows the recovery of a significant number of altimeter measurements, ensuring the continuity and consistency of the correction in the open-ocean / coastal transition zone and also at high latitudes. Figure 4 illustrates the



improved performance of the GPD+ correction over that from ERA-Interim and the composite correction present in the AVISO products.

### 3.3 Ionospheric correction

Within the SLOOP project (Faugere et al., 2010), there has been considerable effort to develop an improved ionospheric correction using an iterative filtering scheme applied to the dual-frequency altimeter missions (TOPEX, Jason-1, Jason-2 and Envisat). This has been independently evaluated by a round robin comparison with previous ionospheric corrections, and it was found that the SLOOP set of corrections led to an improvement in the recovery of mesoscale signals and increased data gain (due to less flagging of suspect data).

For the missions that do not have a second frequency (including Envisat after the loss of S-band data), a model is required. The one used in SL_cci v2.0 is GIM (Iijima et al., 1999), which is based on measurements from GPS satellites. However, prior to 1998 there were insufficient GPS data, so for ERS-1, and ERS-2 we use an interpretation based on the NIC09 climatology (Scharroo and Smith, 2010) modified by contemporaneous TOPEX records of global mean TEC. The corrections for Poseidon are based on the measurements from the DORIS system on-board its satellite.

### 4 Corrections for sea state bias

Sea state bias (SSB) is a correction term encompassing three different effects: electro-magnetic (EM) bias, skewness and tracker bias. A wave field is not usually uniformly covered with identical reflecting facets — the surface tends to be smoother in the troughs of waves than at the crests, so there will be a proportionately stronger response from the lower-lying facets. This effect, the EM bias, will depend upon the radar frequency. Most altimetric retrackers are designed to locate the mid-power point of the leading edge of the waveform; this equates to the median height of reflecting surfaces, rather than the mean. Thus a second effect, the skewness, relates to the difference between the heights of mean and median surfaces, which is a property of the ocean, independent of the radar frequency used for the sensing. The third effect relates to the algorithms used to find the range — this effect will vary with each retracker implemented, but should be the same for identical instruments. However, there are always slight differences between sister instruments e.g. ERS-1 and ERS-2, so the overall sea state bias model is usually determined independently for each altimeter+retracker. In practice, all three of these effects scale roughly with wave height, so the overall sea state bias is expressed as a multiplier of wave height that is a weakly-varying function of sea state conditions.

Although the first two components of SSB should be the same for all Ku-band observing systems, a separate total SSB solution has to be derived for each individual altimeter. For each dataset, minimization procedures are used to express SSB in terms of wave height and wind speed leading to the least variance at crossovers. Many of these solutions remain as defined at the end of their respective missions i.e. once all available data have been analysed. However, as these are



optimizations based on observational data, improvements to the orbits or a change in the modelled PTR or the retracker applied could necessitate a revision to the SSB model.

Early solutions for SSB expressed the SSB coefficient in terms of two key parameters: wave height and wind speed. Those parametric forms are still used for ERS-1 (Gaspar and Ogor, 1994) and Poseidon (Gaspar et al., 1996). A non-parametric

form, offering a better fit to the data, can be achieved for later missions for which there are greater volumes of more precise data. The non-parametric models adopted within SL_cci v2 are for ERS-2 (Mertz et al., 2005), TOPEX (Tran et al., 2010), Jason-1 & 2 (Tran et al., 2012), Envisat (Tran, 2015), GFO (Tran and Labroue, 2009, pers. comm.) and AltiKa (from the PEACHI project). The Cryosat-2 data used in this product are solely those in Low Resolution Mode; at the time that algorithm selection was completed, the most appropriate choice was that derived from Jason-1 GDR-C products, although

ones based on CryoSat-2 data have subsequently become available.

## 5 Corrections for short-term atmospheric and oceanographic phenomena

Our concern within the sea level CCI project is to provide the best dataset for observing climate scale variations in sea level and changes associated with geostrophic currents. Thus short-term effects have to be removed. As these relate to real world physical processes, rather than aspects of the measuring system, these corrections will be expected to be independent of

satellite mission.

### 5.1 Atmospheric pressure correction

Early altimeter processing included an "inverse barometer effect" (IB, see Fu and Pihos, 1994) whereby the sea surface was deemed to be depressed by 1 cm for each increase in atmospheric pressure by 1 mb, with this computed effect being removed from the data to give the sea level expected in the absence of atmospheric effects. Instead a dynamic atmospheric

correction (DAC) was introduced, based on a barotropic global ocean model forced by instantaneous atmospheric pressure and winds fields, and taking into account the ocean dynamic response to atmospheric forcing at high frequencies (Carrere and Lyard 2003) and keeping the IB for low frequencies.

Several atmospheric models have been used to compute the IB and the DAC corrections (ECMWF, ERA-interim, NCEP, JRA-55) in order to find the atmospheric reanalysis most suitable for the present climate analysis. The comparison of input

weather models is another exercise of finding which correction (talking for IB and DAC), when applied to altimeter measurements leads to the greatest consistency between ascending and descending passes and thus reduces the altimeter crossover variance. Figure 5 shows that, for TOPEX/Poseidon data, the JRA-55 model produces greater crossover differences than ERA-interim, with much greater variance in the high southern latitudes where the variability of the atmospheric forcing is strong. Moreover using a DAC forced by ERA-interim significantly reduced the crossover variance

compared with the operational DAC forced by ECMWF analysis (Carrere et al. 2016). Based on such crossover variance analysis, ERA-interim is the preferred model to force the DAC for all missions.





### 5.2 Tides

There are four separate phenomena linked under the label "tides": ocean tides, ocean loading term, solid Earth tide and polar tide. The ocean tide is usually by far the largest, but all four aspects need to be included in order to discern correctly regional variations and long-term trends. An ocean tide model will include many harmonics (not just M2 and S2) and may

be an empirical fit to altimetric sea level data or produced by a high-resolution fluid flow model or a combination of both. Early in the altimetry era there could be as many as 12 independent models to be assessed (Andersen et al., 1995), with, more recently, Stammer et al. (2014) evaluating seven data-constrained models. However, there are presently two main families of solutions to be compared, termed GOT (Ray, 2013) and FES (Carrere et al., 2012). The GOT4.10 solution is mostly based on Jason data excluding those from TOPEX/Poseidon data because of poorly understood effects occurring at

the S2 alias period (59 days). Figure 6 shows that the variance for Envisat data is reduced with the FES2014 model, especially in the Arctic. This model is also very effective in reducing the 59-day signal noted with some of the reference missions (Zawadzki et al., 2017).

The second aspect is the loading tide, which corresponds to the flexing of the Earth in response to the weight of water lying on it. For this we adopt the solution of Ray et al. (2013), which, at the time of algorithm selection, was the only one

consistent with the FES2014 ocean tide. The third aspect is the Earth tide i.e. the changes in the Earth's topography due to the changing gravitational attraction of the moon and sun — here the long established solutions by Cartwright and Tayler (1971), modified by Cartwright and Edden (1973) continue to be applied. Finally, there is the "pole tide", a term describing the small long-period oscillations associated with the movement of the Earth's rotational axis. The recent advance by Desai et al. (2015) takes into account self-gravitation, loading, conservation of mass, and geocentre motion. Moreover this new

model includes a bias and a drift, which means that the new computed pole tide does not include the effects of the Earth's displacement response to that mean pole drift. Removing the long-term mean pole drift has a significant impact on the regional MSL trend estimation; this impact has been validated by comparisons with an Argo database over the time span of the Envisat mission. Thus the recent model of Desai et al. (2015) is the one implemented in SL_cci v2.0.

### 6 Reference surfaces

Since not all missions record measurements over exactly the same points on the Earth's surface, there is a need for an accurate mapped representation of the mean sea surface (MSS) to which observations may be compared (see Eq. 1). Such a 2-D field is frequently revised as new data become available, in particular from CryoSat-2 at high latitudes, and from the 'end of life' geodetic phases of recent missions.

The SL_cci v2.0 is referenced to the DTU15 MSS, and corresponds to a mean over the period 1993-2012. This version is an

improvement on earlier versions (DTU10 & DTU13) in that it makes use of four years of CryoSat-2 data but gives less weighting to data from IceSat (whose large errors gave an unrealistic stripiness to derived MSS fields). Thus the major improvements within DTU15 are the increased data coverage in the high latitudes (both Arctic and Antarctic) and the





Mediterranean, and the finer scales resolved due to the use of shorter correlation scales in the interpolation. Thus, the inter-annual content of the reprocessed v2.0 product will change compared with the previous version due to the evolution of the reference period (1993-2008 for DTU10 in the SL_cci v1.1 product). This will affect assimilating models since these systems are sensitive to the reference period used.

CryoSat-2 data have contributed significantly in the band 82˚-88˚N not sampled by the other radar altimeters, and, due to its long repeat orbit, provides finer longitudinal resolution than the ERS-1, ERS-2 and Envisat instruments for latitudes south of 82˚N. (The DTU15 MSS no longer utilises data from the geodetic phases of ERS-1 and Geosat, as those measurements were noisy; consequently less spatial filtering is required leading to a higher resolution product.) The delay-Doppler mode of CryoSat-2 makes its measurements more resilient to stray reflections from nearby land; thus CryoSat-2 data have led to

marked improvements in the MSS in the Mediterranean and the Bay of Fundy.

### 7 Editing and gridding

The production of the SL_cci v2.0 product uses the same procedures as for the previous version v1 (Ablain et al., 2015). An overview of the different processing steps to produce the Sea Level CCI products can be found in Ablain and Legeais (2014). In brief, these are to acquire and pre-process data, perform input checks and quality control (data are discarded if

flagged for rain, land or ice), inter-calibrate and unify the multi-satellite measurements and generate along-track and gridded merged products.

The unification involves a further orbit adjustment to minimise the difference between tracks and between missions. Then, output checks and quality controls are performed and the multi-satellite along-track data are mapped to generate gridded sea level products. In the processing method, the "reference missions" (TOPEX/Poseidon, Jason-1 and Jason-2) are first

corrected using a sinusoid regression model. These have all been in the same 9.92-day orbital cycle and have a high altitude (1336 km), making their trajectories less sensitive to higher order terms of the Earth's gravity field. Then the "complementary missions" are adjusted to minimise crossovers with the reference orbits and they contribute to increase the spatial resolution of the grids and to increase their accuracy. Thus, the reference missions are used to ensure the stability of the ECV and they determine the global mean sea level and the large-scale changes, but all altimeters contribute to the

detailed SLA patterns. This adjustment towards the orbits of the reference missions also overcomes a poorly understood SLA drift during Envisat's first year of operation.

Finally, output checks and quality control are performed and the multi satellite along-track data are mapped to generate gridded sea level products. In the processing method, reference missions (TOPEX/Poseidon, Jason-1 and Jason-2) are used to ensure the stability of the ECV. The global MSL estimation relies on these reference missions. Other complementary

missions are adjusted on the reference missions and contribute to increase the spatial resolution of the grids and to increase their accuracy. The sensitivity of the gridded products to the mapping algorithms is described in details in Pujol (2012). Different mapping methods were tested in order to assess their ability to accurately reproduce climate signals. This



evaluation has been carried out separating the different temporal and spatial scales related to climate applications. A monthly optimal interpolation is applied (including additional weighted information from part of the previous and following months) to produce maps of sea level on a 0.25˚ grid for the middle of each month. Note that this approach differs from the one used in the production of the DUACS dataset (Pujol et al., 2016) (daily optimal interpolation with different parameters) as the

SL_cci approach has been designed to better answer the needs of climate users.

**8 Data availability**

The gridded monthly files of sea level anomaly at 0.25˚ resolution (DOI: 10.5270/esa-sea_level_cci-MSLA-1993_2015-v_2.0-201612) are freely available (upon email application to info-sealevel@esa-sealevel-cci.org). The Sea Level CCI website (http://www.esa-sealevel-cci.org/products) also contains derived products suitable for some climate studies:

Global Mean Sea Level temporal evolution (DOI: 10.5270/esa-sea_level_cci-IND_MSL_MERGED-1993_2015-v_2.0-201612)

Regional Mean Sea Level trend (DOI: 10.5270/esa-sea_level_cci-IND_MSLTR_MERGED-1993_2015-v_2.0-201612 )

Amplitude and Phase of annual cycle (DOI: 10.5270/esa-sea_level_cci-IND_MSLAMPH_MERGED-1993_2015-v_2.0-201612)

**9 Conclusions**

During phase 2 of the Sea Level CCI project, the consortium has reappraised all the corrections to be used in the production of v2.0 dataset. In some cases, e.g. Earth tide, there has been no change in the recommended correction; in others, such as the pole tide, a new model has become available that is readily endorsed since it significantly improves the accuracy. For

many other terms, there was a choice of two or three corrections: the project evaluated these through a variety of techniques including minimization of mono-mission crossovers, comparison between different altimeter missions, and validation with in situ data. This paper has documented the choices made (Table 1).

The v2.0 dataset was released in December 2016, with details provided at http://www.esa-sealevel-cci.org/products. This will provide a consistent unbiased estimate of sea level spanning 1993-2015, which should greatly enhance the potential for

climatic studies of sea level. The SL_cci ECV v2.0 and validation results are described in Legeais et al. (2017). In terms of the GMSL trend, the change from v1.1 to v2.0 products has led to changes of order 0.1 mm that persist for many months to years, but not led to a significantly different long-term trend (see Fig. 7). The changes that have had the most impact on derived trends are those for orbits and for wet tropospheric correction. Improvements to the time-variable gravity model have lead to major changes in the regional mean sea level trends (>0.5 mm yr$^{-1}$). Through its revision of the calibration of



the MWR on altimetric satellites, the GPD+ solution has a significant impact on the trend of GMSL during the first and second decades of continuous altimetry: -0.2 mm yr$^{-1}$ during 1993-2001 and +0.2 mm yr$^{-1}$ during 2002-2014.

## 10 Author contribution

Phase 2 of the Sea Level CCI project was managed by JFL, who oversaw the evaluation and selection of corrections. The initial draft of the paper was written by GQ. All other authors contributed through their work to evaluate the suite of possible corrections, provision of figures and revision of the text.

## 11 Acknowledgements

The authors acknowledge the support of ESA in the frame of the Sea Level CCI project, launched and co-ordinated by technical officer Jérôme Benveniste.

## 12 Disclaimer

The authors declare that they have no conflict of interest.

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

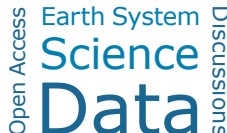

Table & Figures

**Table 1: Summary of the data sources and the corrections applied to each altimeter instrument.**

|  | TOPEX | Poseidon | Jason-1 | Jason-2 | ERS-1 | ERS-2 | Envisat | AltiKa | GFO | CryoSat-2 |
|---|---|---|---|---|---|---|---|---|---|---|
| **Orbit** | GSFC std1504 | | CNES POE-E | | GFZv11 (Rudenko et al., 2017) | | | CNES POE-E | GSFC std08 | CNES POE-E |
| **Data source (Retracker)** | RGDR (least squares) | MLE-3 | GDR-E (MLE-4) | | REAPER | | GDR (Ocean-1) | GDR (Ocean-3) | on-board α-β | GDR (SAMOSA 2.5.0) |
| **Dry Trop** | ERA-Interim | | | | | | | | | |
| **Wet Trop** | GPD+ | | | | | | | | MWR / ECMWF | GPD+ |
| **Iono** | SLOOP | DORIS | SLOOP | | NIC09 | NIC09 / GIM | SLOOP / GIM | GIM | | |
| **SSB** | Tran (2010) | Gaspar et al. (1996) | Tran (2012) | | Gaspar & Ogor (1994) | Mertz (2005) | Tran (2015) | PEACHI | Tran & Labroue (2009) | Tran (2012) |
| **DAC** | ERA-Interim | | | | | | | | | |
| **Ocean tide** | FES2014 | | | | | | | | | |
| **Loading tide** | GOT4v8AC | | | | | | | | | |
| **Earth tide** | Cartwright-Tayler-Edden | | | | | | | | | |
| **Pole tide** | Desai et al. (2015) | | | | | | | | | |
| **MSS** | DTU MSS 2015 | | | | | | | | | |

5  GDR is the Geophysical Data Record, which is the standard product providing altimeter data, with some recommended corrections.



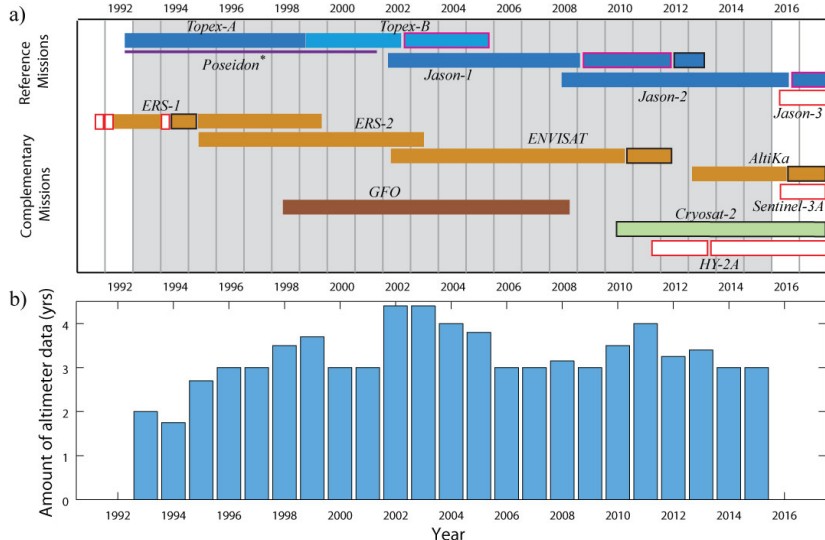

**Figure 1: (a)** Gantt chart of the available altimetry missions. **\*** The spacecraft TOPEX/Poseidon had 2 separate altimeters, with the experimental Poseidon instrument on for ~10% of the time, during which TOPEX did not operate. The "reference missions" all commenced in the same orbit, with 66° orbit inclination and a repeat period of 9.92 days; subsequent phases of those missions

5 were then in a 9.92-day interleaved orbit (pink outline) or a long-repeat (geodetic) orbit (black outline). The missions highlighted in orange were principally in another common orbit (98.5° inclination and 35-day repeat), except for geodetic phases (black outline) and short periods in a 3-day repeat (ERS-1). The other complementary missions are GFO (72° inclination, 17.05-day) and CryoSat-2 (88° inclination, geodetic orbit). The periods indicated by white bars, with red outlines are not used in the production of CCI v2.0 product, **(b)** Annual amount of independent altimeter data used in the production of the v2.0 dataset. (Note, for

10 example, that during the 6-month "tandem" phases between successive "reference missions" the contribution of one of the pair to the sea level record is redundant.



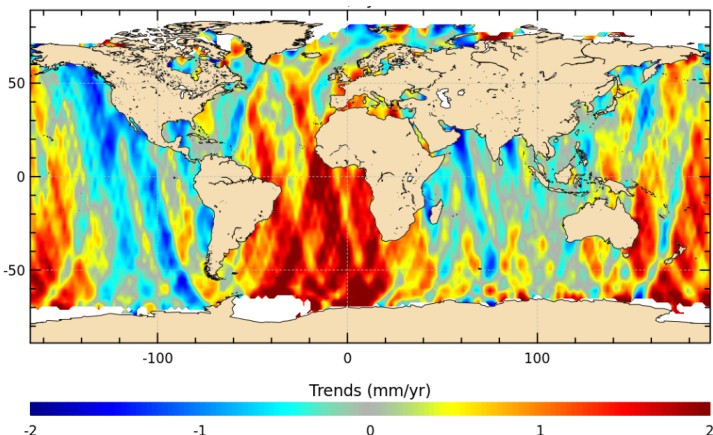

**Figure 2: Difference in sea level trends for ERS-1 data (Oct. 1992 to Jun. 1996) computed using GFZ VER11 orbit and the REAPER combined orbit (which was used in an earlier CCI sea level product (Ablain et al., 2015.**

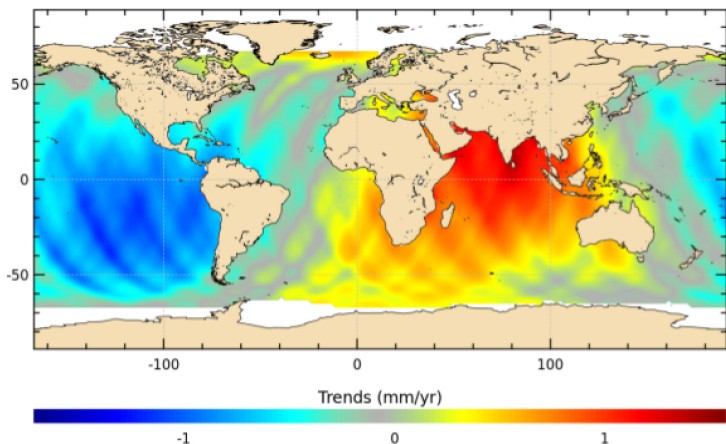

**Figure 3: Change in SLA trend for Jason-1 sea level upon a switch from CNES orbit POE-D to POE-E.**



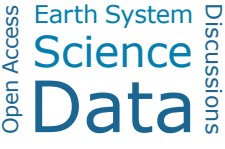

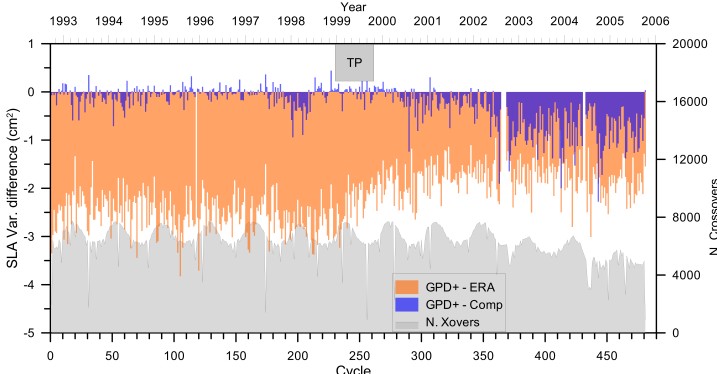

**Figure 4: Difference in variance at TOPEX/Poseidon crossovers for SLA calculated with different WTC. Orange compares GPD+ with ERA-Interim and purple with the composite WTC. Negative values indicate an improvement (i.e. reduction) in crossovers for GPD+.**

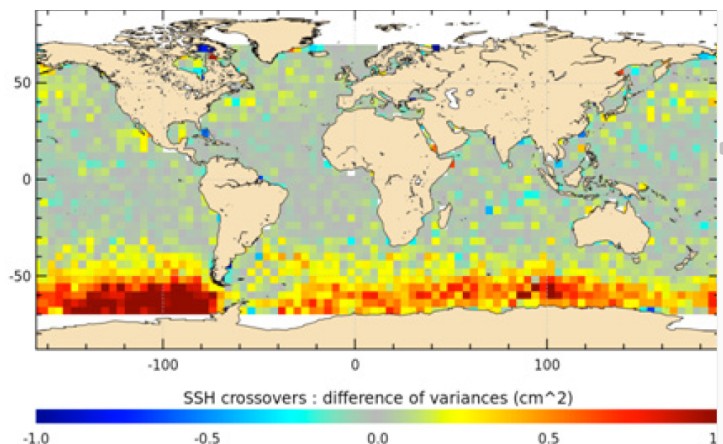

**Figure 5: Change in crossover differences between processing TOPEX/Poseidon mission with IB calculated using JRA-55 or ERA-Interim. Positive values indicate greater variance with corrections from JRA-55.**

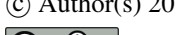


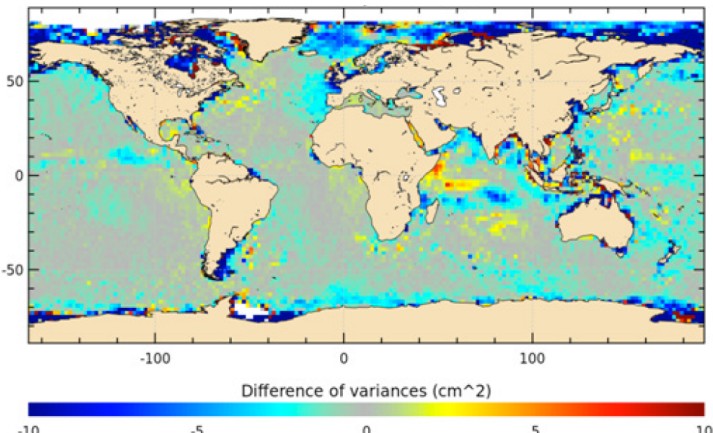

**Figure 6: Change in crossover differences between processing Envisat mission with FES2014 tides or GOT4.8. Negative values indicate reduced variance with FES2014.**

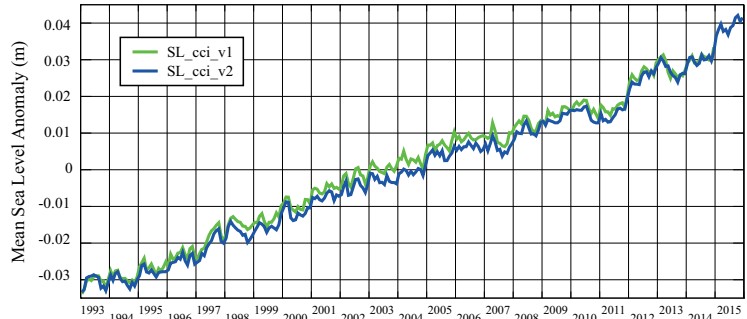

**Figure 7: Comparison of time series of global mean sea level (seasonal signal removed).The v1.1 dataset had been updated til the end of 2014, and has a mean trend of 3.18 mm yr⁻¹); the v2.0, described in this paper, now extends to end of 2015, and has a trend of 3.21 mm yr⁻¹ over the same period as the v1.1.**