# Peer review of "A new phase in the production of quality-controlled sea level data"

_Earth System Science Data, 2017_

## Referee Comment (RC1) · Anonymous Referee #1 · 11 May 2017

This manuscript describes the second version of sea level dataset produced by the European Space Agency's Climate Change Initiative (CCI). The paper summarizes the changes to the dataset from version 1.1. It is suitable for publication in Earth System Science Data as a regular article in the data section. However, as an incremental improvement to an existing dataset, it does not include significant novel concepts or data sources. The length of the dataset has been extended and two new altimeters have been included (SARAL/AltiKa and CryoSat-2). The orbits and all of the instrument and geophysical corrections have been re-evaluated in an effort to create a single gridded dataset of sea level that can be used as a consistent and stable long-term record for use at spatial scales ranging from mesoscale to global. In some cases (e.g. wet troposphere), a new type of correction has been included in this version.

In general, the manuscript provides only a qualitative summary of the evaluations used

for this version of SL_cci (e.g. "significantly reduced the crossover variance"; "shows that the variance for Envisat data is reduced") and much of the improvements are demonstrated in the figures rather than in detail in the text. Without a numerical context, it is not clear from this paper if some of the selections were significant choices or not. For example, on page 5 lines to 8, the evaluation criteria for orbits are described. The next sentence lists the best performing orbits based on these criteria with no quantitative results. I don't understand why this paper does not offer more detailed results, or at a minimum, it could cite specific CCI documentation that provides these details.

Minor comments

Page 4, lines 7 to 9: Add the phrase "when available" or otherwise clarify that some orbit determination instruments are/were not available on all missions.

Page 4, line 3: Insert "to the ocean surface" at the end of the sentence.

Page 4, line 19-20: Some of this sentence is awkward and should be rewritten. "This concept" does not refer directly to anything from the previous sentence. Furthermore, "referred to as" can be dropped, because time-varying gravity was introduced in the previous sentence. The verb "accepts" is not appropriate. I would suggest this rewording:

Detailed analysis of the observations of satellites in low Earth orbit, in particular, from the missions designed to observe Earth's gravity field, such as CHAMP (2000-2010), GRACE (2002-present), and GOCE (2009-2013), has significantly improved knowledge about Earth's static and time-variable gravity. Time variations of the gravity field include the mass redistribution in Earth's atmosphere, hydrosphere, ocean, and cryosphere seasonally and with epoch.

Page 4, line 25: Drop the phrase "to take into account"

Page 4, line 28: Please provide a citation for the GSFC orbits.

Page 6, line 2: Please clarify that the "artefacts" result from the degradation of the

point-target-response of TOPEX-A until 1999 and provide a citation.

Page 7, lines 15 to 16: On my first reading, I didn't understand if the "greater mesoscale signal" was in the dry troposphere correction or in the resulting sea level anomalies, which was not clear. In Section 5.1, page 9 lines 23 to 31 better explain how the atmospheric models were evaluated for the atmospheric pressure correction. Can more of this level of detail be included here?

Page 7, lines 22 to 23. I don't understand why this sentence includes the detail that 10-day missions were adjusted "before" 35-day missions. Aren't these adjustments independent? Does it matter that one set of satellites was adjusted first? Please better explain this adjustment procedure, or drop this sentence.

Page 9 lines 3 to 8: Can the authors clarify which of these sea state bias models were applied in Version 1.1 and which are new to Version 2?

Page 9, line 13. "Thus short-term effects have to be removed." It would be helpful if this section were expanded. Perhaps the authors could briefly explain that short scale temporal variability results in an aliasing problem.

Page 10, line 8: This part of the text identifies GOT4.10 as the latest and best iteration of the GOT tide model. However, Figure 6 uses GOT4.8 in comparison with the FES 2014 without any discussion of GOT4.8.

Page 10, lines 21 to 23: Please either provide a published citation for this result or drop this sentence.
* * *

---

## Referee Comment (RC2) · N. Picot (Referee) · 15 May 2017

[referee-annotated manuscript omitted]

---

## Author Comment (AC1) · 3 Jul 2017

Response to comments on
"A new phase in the production of quality-controlled sea level data"
by Graham D. Quartly et al.

Comments are reproduced verbatim in black, responses in blue, and descriptions of resultant changes to manuscript in black italic.

**Anonymous Referee #1**

This manuscript describes the second version of sea level dataset produced by the European Space Agency's Climate Change Initiative (CCI). The paper summarizes the changes to the dataset from version 1.1. It is suitable for publication in Earth System Science Data as a regular article in the data section. However, as an incremental improvement to an existing dataset, it does not include significant novel concepts or data sources. The length of the dataset has been extended and two new altimeters have been included (SARAL/AltiKa and CryoSat-2). The orbits and all of the instrument and geophysical corrections have been re-evaluated in an effort to create a single gridded dataset of sea level that can be used as a consistent and stable long-term record for use at spatial scales ranging from mesoscale to global. In some cases (e.g. wet troposphere), a new type of correction has been included in this version.
In general, the manuscript provides only a qualitative summary of the evaluations used for this version of SL_cci (e.g. "significantly reduced the crossover variance"; "shows that the variance for Envisat data is reduced") and much of the improvements are demonstrated in the figures rather than in detail in the text. Without a numerical context, it is not clear from this paper if some of the selections were significant choices or not. For example, on page 5 lines to 8, the evaluation criteria for orbits are described. The next sentence lists the best performing orbits based on these criteria with no quantitative results. I don't understand why this paper does not offer more detailed results, or at a minimum, it could cite specific CCI documentation that provides these details.
All the corrections were chosen at a "selection meeting" where the main available options ones were quantitatively compared. In this paper summarising the decisions, we initially tried to make it a "brief communication" by minimising the citation of unrefereed reports, but acknowledge that some readers would like to have visibility of them, and a clearer traceability of the statements in the paper. Consequently the relevant reports (open access on the CCI server) are now cited, along with further documentation on "Topex artefacts", (p.8 l.19-24) as requested. We have also added some numbers illustrating the level of improvements, for example, reduction of the sea surface height crossover standard deviations and mean values, when using new orbits in Section 2.1.
*Many citations and references added.*

Minor comments
Page 4, lines 7 to 9: Add the phrase "when available" or otherwise clarify that some orbit determination instruments are/were not available on all missions.
Agreed that not all sources are available for all altimetric satellites.
*Sentence reworded, adding "although not all sources are available for every satellite"(p.5 l.25-26).*

Page 4, line 3: Insert "to the ocean surface" at the end of the sentence.
Agreed.
*Phrase added (p.5 l.19)*

Page 4, line 19-20: Some of this sentence is awkward and should be rewritten. "This concept" does not refer directly to anything from the previous sentence. Furthermore, "referred to as" can be dropped, because time-varying gravity was introduced in the

previous sentence. The verb "accepts" is not appropriate. I would suggest this rewording:

Detailed analysis of the observations of satellites in low Earth orbit, in particular, from the missions designed to observe Earth's gravity field, such as CHAMP (2000-2010), GRACE (2002-present), and GOCE (2009-2013), has significantly improved knowledge about Earth's static and time-variable gravity. Time variations of the gravity field include the mass redistribution in Earth's atmosphere, hydrosphere, ocean, and cryosphere seasonally and with epoch.

Agreed

*Sentences reworded as "Detailed analysis of the observations of satellites in low Earth orbit, in particular, from the missions designed to observe the Earth's gravity field, such as CHAMP (2000-2010), GRACE (2002-present) and GOCE (2009-2013) has significantly improved knowledge about the Earth's static and time-variable gravity. Time variations of the gravity field include the mass redistribution within and between the Earth's atmosphere, hydrosphere, ocean and cryosphere, on a variety of time scales, from subseasonal to multidecadal." (p.6, l.9-14).*

Page 4, line 25: Drop the phrase "to take into account"

Agreed

*Phrase dropped (no longer in p.6 l.20)*

Page 4, line 28: Please provide a citation for the GSFC orbits.

Only suitable reference is from OSTST presentations

*Lemoine reference added (p.6, l.24). Also on p.7 l. 15-19 we added "Consequently, using the GSFC std1504 orbit for TOPEX/Poseidon instead of the GSFC std1204 orbit (used for the SL_cci v1.0 product) reduces the mean of sea surface height (SSH) crossovers from 0.34 to 0.24 cm. The standard deviation of these crossovers shows an improvement from 4.99 to 4.96 cm for Jason-1, from 4.91 to 4.87 cm for Jason-2, and, from 5.55 to 5.51 cm for Cryosat-2, when using the CNES POE-E orbit instead of the CNES POE-D orbit."*

Page 6, line 2: Please clarify that the "artefacts" result from the degradation of the point-target-response of TOPEX-A until 1999 and provide a citation.

Here we were referring not only to the PTR degradation, but the sawtooth features on the waveform, the energy leakage that shifts with approaching/receding from the Earth, and the averaging of bins within the waveform prior to telemetry

*A few sentences have been added to summarise the complications of TOPEX processing (p.8 l.19-24)*

Page 7, lines 15 to 16: On my first reading, I didn't understand if the "greater mesoscale signal" was in the dry troposphere correction or in the resulting sea level anomalies, which was not clear. In Section 5.1, page 9 lines 23 to 31 better explain how the atmospheric models were evaluated for the atmospheric pressure correction. Can more of this level of detail be included here?

The "mesoscale signal" referred to the SLA variance after atmospheric corrections were applied. However, we recognise that this was not fully clear and so have reworded the text.

*Text amended to read: "... whole 20+ year period, thus the atmospheric model reanalyses are better suited for the present climate purpose. The ERA-interim correction led to a smaller variance of crossover differences than when using the JRA-55 model, especially at southern latitudes where the pressure variability is higher, which indicates a better performance for the ERA-interim model. Thus, considering the long-period reanalyses for climate purposes, the ERA-Interim corrections were the ones adopted for SL_cci v2.0 for all altimeter instruments." (p.11, l.3-8)*

Page 7, lines 22 to 23. I don't understand why this sentence includes the detail that 10-day missions were adjusted "before" 35-day missions. Aren't these adjustments independent? Does it matter that one set of satellites was adjusted first? Please better explain this adjustment procedure, or drop this sentence.

The reviewer is correct: the application of the corrections to each altimeter is independent of the others. It was the determination of the corrections that was done in a specific order because the sun-synchronous altimeters have few crossovers with SSM/I (in a different sun-synchronous orbit), which provide the reference for the technique.

Our aim is to adjust each altimeter mission to the set of SSMI/SSMIS sensors using match points between the two sets of sensors. This can be done between the reference altimetric missions and SSMI/SSMIS because the first are on non-sun synchronous orbits and the second ones are in sun-synchronous orbits. This produces a significant, although time-varying number of matching points. However, because the 35-day missions are sun-synchronous, they are often out of phase with the satellites carrying the SSMI/SSMIS. Thus, the 35-day missions are adjusted to the reference missions, using crossovers and after the first adjustment referred above. In this way, the 35-day missions are also adjusted to the SSMI/SSMIS data set.

We agree that it is easier to leave out this level of detail.

*Sentence dropped (no longer on p.11 l.15).*

Page 9 lines 3 to 8: Can the authors clarify which of these sea state bias models were applied in Version 1.1 and which are new to Version 2?

Yes, the Tran papers in 2012 & 2015 represent the changes with respect to SL_cci v1.1.

*Text added to specify changes since SL_cci v1.1 (p.13, l.19-20)*

Page 9, line 13. "Thus short-term effects have to be removed." It would be helpful if this section were expanded. Perhaps the authors could briefly explain that short scale temporal variability results in an aliasing problem.

The intention is to provide a monthly dataset suitable for studying long-term variations, thus the effect of sub-monthly variations should be minimised as they will not be adequately resolved in time.

*Revised text: "The temporal sampling by altimeters is insufficient to resolve all time-scales, so high-frequency ocean variability is aliased to longer time scales, thus polluting climate estimations if not adequately corrected. Thus, short-term effects have to be removed using accurate physical ocean models, which are expected to be independent of satellite missions." (p.13 l.23-27)*

Page 10, line 8: This part of the text identifies GOT4.10 as the latest and best iteration of the GOT tide model. However, Figure 6 uses GOT4.8 in comparison with the FES 2014 without any discussion of GOT4.8.

This was an error: the plot using GOT4.10 was substituted shortly before submission, but the caption was not changed.

*Caption changed to state GOT4.10.*

Page 10, lines 21 to 23: Please either provide a published citation for this result or drop this sentence.

This comment was about the choice of pole tide model. The current paper is about the process of calculating a self-consistent record of global sea level; a later paper will describe the assessment of the quality of this new dataset; we now add references to the original selection meeting and to that paper in preparation.

*Added text: "(ASM, 2015a; Legeais et al., 2017)". (p.15, l.19)*

**Review of Nicolas Picot**
I recommend publication in Earth System Science Data as a regular article in the data section

(Detailed comments are extracted from annotated manuscript, and given page and line numbers by the authors.)

p.1 l.33-34  I found this last sentence a bit unfortunate (even if true …) If the GMSL trend is not modified overall there is a lot of improvements obtained. I encourage the author to clarify this point in the introduction.
Fig. 7 shows there has been a reduction in sea level estimates for 1999-2011 compared with v1.1, but more recent periods match between the two datasets.  Thus the overall trend is largely unchanged, although the rate of sea level rise was lower in first decade and higher in the second one.
*Text has been added to the end of abstract that "there is now greater confidence in this result as the errors associated with several of the corrections have been reduced.  Compared with v1.1 of the SL_cci dataset, the new rate of change is 0.2 mm yr$^{-1}$ less during 1993 to 2001 and 0.2 mm yr$^{-1}$ higher during 2002 to 2014." (p.2 l.11-14)*

Also there is recent analysis of TopexA data (refer to A. Cazenave paper) that would be worth to mention.
Clearly the current paper is about the corrections and processing that were used to make the SL_cci v2.0, and do not reflect what is "cutting edge" in terms of research, but that which had been developed and independently assessed by the CCI team by Nov. 2015  (in order to be used consistently throughout the processing of the data that accompanies this article).  We thank the reviewer for pointing out the article on Topex A (Dieng et al.);
*We now cite Dieng et al. (2017), Watson et al. (2015) and Chen et al. (2017) as part of the still ongoing research to improve sea level records (p.8, l.25-27).*

p.2 l.15  add the GIA (glacial isostatic adjustment) in the list.
Concept is now mentioned.
*Text added: " Also, the land masses are still undergoing a delayed response to the removal of their burden from the last ice age (a phenomenon known as "glacial isostatic adjustment")." (p.3, l.7-8)*

p.3 l.6 'MSS'   For Topex/JAx or ERS/ENVISAT we are more using the Mean Profiles than the MSS.
The SL_cci SLAs have been produced using a Mean Sea Surface for all missions. No mean SSH profiles have been used (contrary to CMEMS products).  This is because we focussed on ensuring the stability of the sea level record and wanted to avoid potential bias.
*No change to text.*

p.7 l.5-7  But still not fulfilling the accuracy requirement for the reference missions.
We agree with the reviewer's comment (about ionospheric models).
*Text added " and indeed the measuring and modelling of the ionospheric correction is a topic that still needs further development" (p.10 l.19-20)*

p.7.l.15-16   To be clarified, this might be confusing for the end user. An increase of mesoscale signal can be a good thing.
Assuming that the errors (especially atmospheric delays) are uncorrelated with the SLA signal, total variance equals sum of variance of signal plus variance of each of the errors.  Thus, a common approach is to look for a model that minimises the observed variance, as that should have the smallest errors.
*Text rewritten as: "The ERA-interim correction led to a smaller variance of crossover differences than when using the JRA-55 model, especially at southern latitudes where the pressure variability is higher, which indicates a better performance for the ERA-interim model." (p.11, l.4-6)*

p.7 l.32  Please add some text to explain why the composite correction seems to be closer to the GPD+ correction after year 2003.

The figure shows exactly the opposite. As discussed in Fernandes et al, 2015, the results are significantly different in the first and second part of the mission. Partly, this is likely to be due to anomalies detected in the tape recorder of the TMR, occurred at cycle 370, which caused measurement gaps. The interpolation of these missing measurements originated spurious values that are mostly spotted and corrected by the GPD algorithm, but not by the Composite correction.
No change

p.7, l.32 GPD+ is always better than the composite one? (Fig4 might be confusing on this issue).
No this is not true: Variance with GPD+ minus Variance with Comp is occasionally positive, indicating that there are a few cycles for which the Comp correction gives the lesser variance.
No change

p.8 l.11  Some work are ongoing on JPL side to try to deliver a GPS based Iono correction for the whole Topex period (so starting in 1993).
We thank the reviewer for his comment on recent developments.  We felt that the GPS observations were "insufficient" at the time that algorithms/corrections had to be specified for the SLcci_v2.0 product; we look forward to evaluating any forthcoming GPS-based product for the early 1990s, when it becomes available.
*We have changed "insufficient" to "relatively few" (p.12, l.10)*

p.8 l.25   correct but the effect of roughness/wind and swell cannot be ignored. Was the 3parameter method developed by Ngan envisaged in the frame of this project ?
The statement in the paper is correct: it does vary roughly with wave height, but roughness, wind and swell affect the proportionality.  We state clearly which algorithms by Ngan Tran were used, and now also what the change is from v1.1.  Thus the paper clearly states what has been used in the production of the available product, rather than some "3-parameter algorithm" that was not evaluated at the time of selection, and thus not used.
*No changes to text.*

p.9 l.10  But not used in the frame of this project ?   Why ?
Because the models (for Cryosat SSB) were not available for assessment at the time decisions had to be made on a processing strategy.
*No change.*

p.9 l.13   Might be confusing for end users. Please clarify.
The intention is to provide a monthly dataset suitable for studying long-term variations, thus the effect of sub-monthly variations should be minimised as they will not be adequately resolved in time.
*Revised text: "The temporal sampling by altimeters is insufficient to resolve all time-scales, so high-frequency ocean variability is aliased to longer time scales, thus polluting climate estimations if not adequately corrected.  Thus, short-term effects have to be removed using accurate physical ocean models, which are expected to be independent of satellite missions." (p.13, l.23-27).*

p.10 l.2  Might be worth to add the 'Internal Tide' in the list - this topic is becoming more and more important.
Agreed.
*"Internal tide" is now mentioned in the list at the beginning of this section (p.14, l.22), and the need for an adequate model is referred to at the end of the section (p.15 l.21-22).*

p.10 l.29  What is the rationale beyond this selection ?
At the time of model selection MSS CNES_CLS2015 was not available for evaluation.  Thus the DTU series was preferable (for its coverage of the high latitudes), with DTU15 offering a number of advantages (in terms of data used) compared with prior versions.

*Text added referring to the coverage of high-latitude Arctic Ocean (p.16, l.2), as well as the reference to "finer spatial scales" already in the text (p.16, l.8).*

p.11 l.20  Might be worth to describe this with more details (how many orbits, order of magnitude, ...)
We have expanded the explanation to discuss first of all the adjustment of the reference missions, and then the unification of the complementary missions through minimisation of crossovers.  We have added a reference (Le Traon & Ogor, 1998) for the method, but thought that further detail on this specific issue was beyond what the reader would want.
*Paragraph fully revised (p.16, l.26 - p.17,l.9))*

p. 11 l.21  And to the drag effects.
Agreed
*Briefly mentioned on p.17, l.5).*

p.11 l.25 Indeed not understood at all ...
Agreed.
*Words "poorly understood" replaced with "spurious" (p.17, l.10)*

p.11 l.29-31   The same text is just above. 'Then the "complementary missions" are adjusted to minimise crossovers with the reference orbits and they contribute to increase the spatial resolution of the grids and to increase their accuracy.'
We apologise for this error in moving text by copying and pasting that led to a duplication
*The redundant text had been removed (no longer on p.17, l.13).*